# Circulating Biomarkers of Response and Toxicity of Immunotherapy in Advanced Non-Small Cell Lung Cancer (NSCLC): A Comprehensive Review

**DOI:** 10.3390/cancers13081794

**Published:** 2021-04-09

**Authors:** Alice Indini, Erika Rijavec, Francesco Grossi

**Affiliations:** 1Medical Oncology Unit, Fondazione IRCCS Ca’ Granda Ospedale Maggiore Policlinico, 20122 Milan, Italy; alice.indini@policlinico.mi.it; 2Unit of Medical Oncology, Department of Medicine and Surgery, University of Insubria, ASST dei Sette Laghi, 21100 Varese, Italy; Francesco.grossi@asst-settelaghi.it

**Keywords:** NSCLC, PD-L1, ctDNA, CTCS, miRNAs, exosomes, immunotherapy

## Abstract

**Simple Summary:**

Although immunotherapy has dramatically revolutionized non-small cell lung cancer (NSCLC) treatment, not all the patients will benefit from this innovative therapy. The identification of potential biomarkers able to predict efficacy and toxicity of immunotherapy represents an urgent need for tailored treatment regimens. Liquid biopsy is a minimally invasive and economical tool that could provide important information about patients’ selection and treatment monitoring. Currently, several blood biomarkers are under investigation (circulating immune and tumor cells, soluble immunological mediators, peripheral blood cells). Prospective clinical trials are needed to validate their use in clinical practice.

**Abstract:**

Immune checkpoint inhibitors (ICIs) targeting the programmed cell death (PD)-1 protein and its ligand, PD-L1, and cytotoxic T-lymphocyte-associated antigen (CTLA)-4, have revolutionized the management of patients with advanced non-small cell lung cancer (NSCLC). Unfortunately, only a small portion of NSCLC patients respond to these agents. Furthermore, although immunotherapy is usually well tolerated, some patients experience severe immune-related adverse events (irAEs). Liquid biopsy is a non-invasive diagnostic procedure involving the isolation of circulating biomarkers, such as circulating tumor cells (CTC), cell-free DNA (cfDNA), and microRNAs (miRNAs). Thanks to recent advances in technologies, such as next-generation sequencing (NGS) and digital polymerase chain reaction (dPCR), liquid biopsy has become a useful tool to provide baseline information on the tumor, and to monitor response to treatments. This review highlights the potential role of liquid biomarkers in the selection of NSCLC patients who could respond to immunotherapy, and in the identification of patients who are most likely to experience irAEs, in order to guide improvements in care.

## 1. Introduction

Over recent years, the management of several tumors including non-small cell lung cancer (NSCLC) has been radically improved with the advent of immune checkpoint inhibitors (ICIs). These agents include antibodies targeting the programmed cell death receptor-1 (PD-1) (nivolumab, pembrolizumab, and cemiplimab), the anti-programmed death 1 ligand (PD-L1) (atezolizumab, durvalumab, and avelumab), and the cytotoxic T-lymphocyte antigen 4 (anti-CTLA-4) (ipilimumab, and tremelimumab). To date, several ICIs have been approved for the treatment of NSCLC, both as frontline therapies or after progression to chemotherapy [1,2]. Unfortunately, despite the progress obtained with ICIs, most patients do not benefit from immunotherapy and eventually experience disease progression. Hyperprogressive disease (HPD), a peculiar pattern of rapid acceleration of tumor growth kinetics, has been reported in several malignancies after treatment with ICIs, and is often associated with worse survival outcomes [3].

Several predictive and prognostic biomarkers have been investigated to guide patient selection. Clinical trials have shown a positive association between PD-L1 expression on tumor tissue and tumor mutational burden (TMB), and disease response to immunotherapy [4]. In June 2020, the United States (US) Food and Drug Administration (FDA) approved high TMB as a criterion for pembrolizumab treatment across different tumor types [5]. Currently, PD-L1 expression is the only biomarker used in clinical practice to select NSCLC patients most likely to benefit from ICIs. However, even patients with low PD-L1 expression exhibit response to immunotherapy, thus meaning that a single biomarker may not be indicative for patient selection [6]. Moreover, there are significant inconsistencies across clinical trials, and multiple antibody-staining PD-L1 immunohistochemistry (IHC) assays have been approved as companion diagnostics for ICIs [7]. Assessment of PD-L1 expression can be challenging in NSCLC patients, due to sampling problems (both techniques require adequate tissue biopsies), dynamic changes in PD-L1 expression over time, and intra-tumor spatial heterogeneity, implying that analyzed tumor tissue may not be indicative of the whole tumor microenvironment (TME) [8].

Biomarkers have been also investigated to predict toxicity during immunotherapy. ICIs are well tolerated, and immune-related adverse events (irAEs) are usually easily manageable. However, some toxicities may become life threatening especially if not promptly diagnosed [9]. Results from a meta-analysis of trials investigating ICIs showed that fatal toxicities occurred in 0.3–1.3% of patients, and were more commonly observed in subjects treated with a combination of different ICIs (e.g., anti-CTLA4 and antiPD1) [10]. Several studies demonstrated that the onset of irAEs correlate with better response to immunotherapy [11,12,13]. However, there is also evidence supporting a potential detrimental effect of irAEs on survival outcomes, depending on the type, entity and seriousness of irAEs. A typical example is represented by ICI-related pneumonitis, which represents an uncommon but potentially fatal irAE [14,15]. Thus, there is a complex relationship between the immune system activation against the tumor and its concomitant effect on self-tissues, which deservers further research.

In order to bypass the intrinsic limitations of tissue biomarkers, research has focused on circulating biomarkers predictive of response and toxicity in patients receiving ICIs [16]. The major advantages of the so-called liquid biopsy consist in minimal invasiveness, allowing repeated assessments over treatment, and in the possibility to perform parallel testing of several biomarkers with a single blood sample [17]. Promising results, though, go together with significant technical challenges and limited knowledge in this field. Other intrinsic challenges include the lack of standardized panels and unified cut-offs, and the need for high-quality DNA to perform the analyses. Liquid biopsy has been largely investigated in the setting of targeted therapies for NSCLC, while its role during immunotherapy is still under definition [18,19].

In this review, we provide an overview of available data on circulating biomarkers of response and toxicity of immunotherapy in patients affected by advanced NSCLC, analyzing strengths and weaknesses of these novel methods, and potential implications for future research. A summary of the main results from clinical trials on liquid biopsy in NSCLC patients receiving immunotherapy is described in Table 1. Ongoing clinical trials are reported in Table 2.

## 2. Circulating Tumor DNA (ctDNA)

Small amounts of cell free DNA (cfDNA) can be detected in human plasma at very low concentrations (5–10 ng/mL) under physiological conditions [36]. The amount of cfDNA is significantly increased in cancer patients compared to healthy controls, and is higher in patients with advanced compared to earlier staged tumors, due to the release of DNA fragments in the blood stream, which are called circulating tumor DNA (ctDNA) [37]. There is a high concordance between ctDNA and tumor cells in advanced NSCLC, meaning that ctDNA contains specific mutations identical to those found in the primary tumor and its metastases [38]. Moreover, ctDNA has a relatively short half-life (15 min to 2.5 h), which allows for timely monitoring of tumor mutational status, and its dynamic changes over time. Several studies have demonstrated that the presence of high concentration of cfDNA in patients with advanced NSCLC correlates with poor survival outcomes, irrespectively of histology, clinical characteristics, and systemic treatments [39,40,41]. There is also evidence that NSCLC patients harboring somatic alterations of >5% variant allele frequency experienced worse survival outcomes [42]. Altogether, these elements make ctDNA an appealing biomarker for dynamic assessment of druggable mutations, with the aim to drive treatment selection on the basis of evaluation of circulating genetic targets at the time of disease progression.

There is a wide range of detectable ctDNA amount across different tumor types and stages, ranging from 0.01% to more than 90% of total cfDNA [43]. Several high-sensitivity approaches, including digital PCR (dPCR) and next generation sequencing (NGS) techniques, have been evaluated for the detection of ctDNA in patients with cancer [44]. Specifically, dPCR are used to detect a low number of genes (usually three to five), while NGS techniques allow the detection of multiple single nucleotide variants at the same time (including single-gene fusions, insertions, and/or deletions), with the aid of a single panel. These methods are used to assess both cfDNA and ctDNA levels, and blood TMB (bTMB) [44].

The role of bTMB profiling and ctDNA sequencing in NSCLC patients treated with ICIs has been explored in several studies. The first investigation demonstrating that TMB can be measured in plasma, and that bTMB is associated with clinical benefit from ICI was performed on patients with NSCLC (*n* = 853) treated with first-line atezolizumab in two clinical trials, the POPLAR (NCT01903993) and the OAK trial (NCT02008227). Data from this analysis demonstrated that progression free survival (PFS) was longer in patients with combined high PD-L1 expression on tumor tissue and high bTMB (i.e., ≥16) [20]. In the subsequent study by Wang et al., a 150-cancer genes panel (CGP) named NCC-GP150 was designed and virtually validated using The Cancer Genome Atlas database [21]. The aim of this study was to evaluate the correlation of bTMB in two independent cohorts of patients with advanced NSCLC receiving ICIs (*n* = 98). Beside the validation of the NCC-GP150 panel for ctDNA-based bTMB measure, this study suggests that bTMB can be used as a potential biomarker to estimate response during ICIs, with more significance in first- or second-line, rather than in later-lines of treatment.

The association of ctDNA changes with radiographic responses during ICI was also investigated. Goldberg et al. compared longitudinal variations in ctDNA levels with changes in radiographic tumor size and survival in NSCLC patients treated with ICI (*n* = 28) [22]. In this study, ctDNA was quantified by determining the allele fraction of cancer-associated somatic mutations in plasma, using a multigene NGS assay. Results of this study suggest that the reduction of ctDNA levels to less than half of the baseline value correlated with response and survival, and was also an earlier biomarker of response compared to radiographic assessment. Similarly, the study of Iijima et al. found that plasma analysis of ctDNA performed after the first two weeks of treatment, further validated by tumor tissue analysis, could predict a durable response to nivolumab in NSCLC patients with high tumor burden [23]. Anagnostou et al. demonstrated that patients without an on-treatment drop of ctDNA levels had shorter survival compared with molecular responders: ctDNA could be detected earlier (average 8.7 weeks) and was more reliable than radiographic imaging in predicting clinical benefit from treatment. This study also showed that T cell expansion, measured through increases of T-cell receptor productive frequencies, paralleled ctDNA reduction in patients responding to ICIs [45].

Changes in allelic variant frequencies have been shown to predict response to ICI. Raja et al. found that a decrease in mean allelic variant frequencies was associated with longer treatment duration, better response and survival among patients with NSCLC and urothelial cancer receiving durvalumab in the Study 1108 (NCT01693562) and the ATLANTIC trials (NCT02087423) [24]. Early alterations in ctDNA allele fraction correlated with clinical outcomes during ICI therapy also in the study by Guibert et al. [25].

The intrinsic limitations of the reported research lie in the retrospective nature and the heterogeneity of methods across studies. Currently, several investigations are ongoing to prospectively evaluate the role of bTMB and ctDNA in patients treated with ICIs. Final efficacy analysis from the B-F1RST trial (NCT02848651) showed that bTMB is a predictive biomarker for survival in NSCLC patients receiving first-line atezolizumab [26]. The BFAST trial (NCT03178552) is a phase II/III multi-cohort study designed to evaluate the safety and efficacy of targeted therapies or immunotherapy as single agents or in combination in patients with advanced NSCLC harboring oncogenic somatic mutations or positive by TMB assay, as identified by two blood-based NGS ctDNA assays [46].

## 3. Circulating Tumor Cells (CTCs)

Both the primary tumor and its metastases can release tumor cells into the blood stream, namely circulating tumor cells (CTCs) [47]. A high CTC count and the presence of PD-L1 positive CTCs are independent negative prognostic factors patients with NSCLC receiving ICIs [48]. CTCs are rare (~1 out of 100 million normal blood cells), and their detection can be challenging even in patients with advanced disease [49]. Several techniques have been developed to detect CTCs, but to date there is no consensus regarding which features should be used to confirm the malignant nature of CTCs (e.g., molecular profile, or surface antigen expression). This indeed represents a relevant issue for the identification of CTCs, as it depends on the specificity of isolation techniques [50].

The CellSearch^®^ (Menarini Silicon Biosystems; Florence, Italy) has been the first US FDA approved assay for CTCs’ detection and quantification [51]. This method detects CTCs with the aid of antibodies targeting the anti-epithelial cell adhesion molecule (EpCAM) antigen. One of the main limitations of CellSearch is the variable expression of EpCAM on CTCs, along with technical challenges in CTCs isolation. Isolation by size of epithelial tumor cells (ISET) is another method which allows for the isolation of CTCs by filtration based on their different size compared with normal circulating cells [52]. This method can therefore detect EpCAM negative CTCs, however it would not filtrate very small CTCs well. Other promising techniques to improve CTCs identification rely on morphology analysis, RT-PCR, and detection by telomerase activity and expression [53,54].

Several studies investigated the prognostic and predictive role of CTCs in patients with NSCLC [55], and demonstrated a correlation of CTC count with worse response and survival during treatment with ICIs [27,40]. Recently, Castello et al. evaluated the relationship between CTCs and metabolic 18F-FDG positron emission tomography (PET)-based indexes in 20 NSCLC patients. In this case series, CTCs provided valuable information on tumor metabolic activity, which could be regarded as a surrogate of tumor aggressiveness [28].

Most studies focusing on the role of PD-L1 positive CTCs and prognosis of patients treated with ICIs provided controversial results. In the study by Nicolazzo et al., CTC monitoring at three prespecified timepoints (after 1, 3 and 6 months from the beginning of treatment) during nivolumab treatment, allowed to divide patients with NSCLC into two subgroups according to PD-L1 expression [29]. Patients with PD-L1 negative CTCs experienced clinical benefit from treatment, while those with PD-L1 positive CTCs had disease progression. The persistence of PD-L1 positive CTCs at different timepoints could reflect the presence of resistance mechanisms during treatment. Guibert et al. prospectively assessed the presence of PD-L1 positive CTCs at baseline, and at the time of disease progression in patients with NSCLC undergoing treatment with nivolumab [30]. A high proportion of PD-L1 positive CTCs was associated with poor survival outcomes, and all patients undergoing disease progression had detectable PD-L1 positive CTCs. As already shown in previous investigations, this study confirmed the lack of concordance of PD-L1 positivity on CTCs with tumor tissue [56]. Only one study demonstrated that combined PD-L1 positivity on CTCs and PD-L1 positive staining on tumor tissue could be a biomarker for better response to ICIs in patients with advanced NSCLC, while another study showed no correlation with PFS [31,32].

Overall, evidence suggests that CTC count could be a promising biomarker of benefit during ICI treatment. However, technical issues together with controversies of available results to date require further research implementation before CTCs can be translated into the clinic.

## 4. Circulating MicroRNAs and Exosomes

Along with ctDNA, circulating RNA molecules and exosomes can be found in biological fluids. MicroRNAs (miRNAs) are short, non-coding, single stranded RNA molecules, which regulate tumor suppressor genes and oncogenes expression at the post transcriptional level [57]. Exosomes are small extracellular vesicles containing DNA, mRNAs, non-coding miRNAs, proteins, and lipids, which are released into the extracellular space by normal cells. The amounts of exosomes and miRNAs in the peripheral blood of cancer patients is higher than in healthy controls, and can be detected through RT-PCR and NGS assays [57,58,59]. Moreover, circulating miRNAs mirror the biology of the tumor, making them ideal biomarkers for diagnosis and on-treatment disease monitoring.

To date, only a limited number of small studies have focused on the correlation of exosomes and miRNAs with outcomes during ICIs. Data from a small cohort of melanoma and NSCLC patients treated with nivolumab showed a significant association between exosomal PD-L1 levels and treatment response [33]. In the study by Halvorsen et al., a 7-miRNAs signature was associated with better OS in patients receiving nivolumab [34]. Another study showed an association between the downregulation of circulating miRNA-320b and -375 expression and response to immunotherapy [35]. Interestingly, along with miRNAs, this study investigated also several other soluble mediators and their potential role in predicting treatment related toxicity.

Peng et al., evaluated the role of plasma-derived exosomal miRNAs as biomarkers for selection of patients before ICI treatment [60]. In their study, 30 patients with advanced NSCLC receiving immunotherapy were evaluated. Three miRNAs from hsa-miR-320 family (hsa-miR-320d, hsa-miR-320c, and hsa-miR-320b) were identified as potential predictors, and one as a potential target for anti-PD-1 treatment (hsa-miR-125b-5p). The three miRNAs were all upregulated in patients with progressive disease, and correlated with an unfavorable response to anti-PD-1. On the other hand, hsa-miR-125b-5p was downregulated in patients with response to anti-PD-1 treatment. Based on these results, the authors suggest that continuous monitoring of hsa-miR-125b-5p levels during anti-PD-1 treatment may be taken into consideration as an on-treatment diagnostic tool, especially for the subset of patients showing delayed responses or pseudo-progression.

Another study by Fen et al., demonstrated that alterations in certain circulating miRNAs are associated with response and survival during ICIs [61]. The authors performed a discovery assay with miRNA profiling, showing a different profile of 27 miRNA expression (of which 22 were highly, and 5 were lowly expressed) among 19 NSCLC patients who were responders compared to 27 non-responders to immunotherapy. Subsequently, they validated 10 miRNAs which were significantly highly expressed in an independent cohort of patients. The relevance of this study is that patients with disease response had longitudinally increased, 10 highly expressed miRNA pattern levels from pre-treatment to post-treatment, and this associated with better survival.

Though promising, available data on miRNAs and exosomes are too limited to drive any significant conclusion on their possible role in predicting response and survival during immunotherapy, due to the small sample size of available studies and the different techniques. Integration of miRNAs and exosomes quantification with other circulating biomarkers could be a strategy to overcome the intrinsic limitation of this technique, and to increase the predictive role of single biomarkers.

## 5. Soluble Immunological Mediators

Soluble mediators (including the soluble form of PD-L1 [s-PD-L1], cytokines and chemokines), have been investigated given the potential role of an “immunogenic” signature to predict response and resistance to immunotherapy [62]. Techniques for the assessment of soluble mediators include RT-PCR and immunoassays, such as the enzyme-linked immunosorbent assays (ELISA). High levels of s-PD-L1 were associated with poor clinical benefit, treatment failure and subsequent worse survival outcomes during immunotherapy across different studies [63,64]. Moreover, low baseline s-PD-L1 levels were associated with better treatment outcomes, while an early increase in s-PD-L1 levels during treatment was identified as a negative prognostic factor. As previously observed with PD-L1 positive CTCs, even for s-PD-L1 there is no strict correlation with PD-L1 expression on tumor tissue, again supporting the dynamic changes of this biomarker over time [58]. Costantini et al. evaluated the role of soluble biomarkers in predicting irAEs during nivolumab treatment: sPD-L2, and interferon-gamma (IFN-γ) and soluble interleukin (IL)-2 at different timepoints were associated with severe treatment-related toxicity [33].

Boutsikou et al. investigated the prognostic and predictive role of IFN-γ, tumor necrosis factor-alpha (TNF-α) and a panel of ILs in the peripheral blood of 26 NSCLC patients treated with pembrolizumab or nivolumab. At the time of diagnosis and at three months after initiation of immunotherapy, IFN-γ, TNF-α, IL-1β, IL-2, IL-4, IL-5, IL-6, IL-8, IL-10 and IL-12 were analyzed by flow cytometry. Interestingly, patients with increased IFN-γ, TNF-α, IL-1β, IL-2, IL-4, IL-5, IL-6, IL-8, IL-10 and IL-12 levels showed better response and longer survival with ICIs than patients with lower levels [65].

Oyanagi et al. evaluated the serum proteins levels in 38 NSCLC patients treated with nivolumab, and evaluated their association with tumor response and the onset of irAEs with the aid of a multiplex quantitative protein assay (Milliplex^®^ Multiplex Assays for Luminex^®^; Merck, Darmstadt, Germany). Blood samples were collected at baseline and at week four. The authors demonstrated that the baseline serum levels of IP-10 and follistatin could be potential biomarkers associated with clinical benefit, and that RANTES levels at week four were associated with irAEs’ onset [66].

Several trials investigated the prognostic role of tumor makers. A better OS has been observed in NSCLC patients receiving immunotherapy with a reduction in serum level of carcinoembryonic antigen (CEA) or cytokeratin-19 fragments (CYFRA 21-1) [67,68].

## 6. Peripheral Blood Cells

### 6.1. White Blood Cells’ Count

In several types of solid tumors, including NSCLC, multiple markers of systemic inflammation, which can be detected in peripheral blood, correlate with outcomes of immunotherapy non-specific biomarkers, like the neutrophil to lymphocyte ratio (NLR), calculated by dividing absolute neutrophil counts by lymphocyte counts, and the derived neutrophil to lymphocyte ratio (dNLR), calculated as the baseline lymphocyte count (BLC)/(white blood cell-count BLC), have been the first easy-to-use parameters correlating with outcomes of immunotherapy [69].

Results from a meta-analysis of 17 published studies, involving 2106 advanced NSCLC patients treated with ICIs, showed that high pre-treatment NLR was significantly associated with shorter PFS (*p* < 0.001), and OS (*p* < 0.001) [70].

Results from a retrospective study involving 167 patients with solid tumors (including NSCLC) treated with nivolumab or pembrolizumab, showed that patients with higher baseline lymphocyte counts (ALC > 2000) had an increased risk of irAEs [71].

Pavan et al. investigated the potential role of NLR and platelet-to-lymphocyte ratio (PLR) as predictors of irAEs in 184 advanced NSCLC patients treated with ICIs. Patients with low NLR and PLR at baseline had a higher rate of irAEs [72].

### 6.2. Circulating Immune-Suppressive Cells

Flow cytometry (FC) allows a deeper analysis of peripheral blood cells subpopulations, including monocytic myeloid-derived suppressor cells (M-MDSCs), granulocytic MDSC (G-MDSCs), natural killers (NK), and regulatory T cells (Tregs). The resident fraction of MDSCs play an immunosuppressive role within the tumor tissue, while their circulating counterpart exert a negative effect on anti-PD1 therapy in animal tumor models [73]. A rapid increase in NK cell fraction was detected in patients with NSCLC responding to nivolumab treatment, along with a reduction of a specific subset of G-MDSCs [74]. The NK cell to G-MDSC ratio (NMR) was significantly higher in responders compared with non-responders, and correlated with response and survival outcomes. Research has found a correlation on the phenotype of peripheral blood T cells and clinical outcomes of ICI treatment. Indeed, not only is PD-1 expression on CD4^+^ T cells significantly elevated in NSCLC patients compared with healthy subjects, but there is also a strong association between high expression of PD-1 on CD4^+^ T cells and poor survival outcomes [75,76]. Biomarkers of exhaustion undergo dynamic changes over treatment. In the study by Kamphorst et al., the majority of patients with disease progression had either a delayed or absent PD-1+ CD8^+^ T cell response, while most patients with clinical benefit showed increased PD-1+ CD8^+^ T cell responses within the first month of treatment [77]. Ottonello et al. observed higher levels of CD3^+^, CD4^+^, and CD8^+^ T cells, but lower levels of NK cells and PD-1 positive CD3^+^ and CD8^+^ T cells at baseline in patients who experienced longer OS with nivolumab. Furthermore, the authors showed that patients with progressive disease had significantly higher levels of exhausted T cells compared with patients with disease response [78].

Mazzaschi et al. showed that the integrated analysis of sPD-L1, CD8+PD-1+ and NK cells and blood descriptors of the inflammatory response (i.e., dNLR, and lactate dehydrogenase (LDH)) could be combined in a prognostic and predictive immune score, in a cohort of 109 advanced NSCLC treated with ICIs [79,80]. Prelaj et al. validated a prognostic score (the EPSILoN score) based on blood parameters (LDH, NLR) and clinical characteristics (Eastern Cooperative Oncology Group (ECOG) performance status (PS), smoking status, and presence of liver metastases) in 193 advanced NSCLC patients receiving immunotherapy as second- or further-line [81].

## 7. Conclusions

The increasing use of immunotherapy in the treatment of NSCLC, especially in patients with early-stage disease, requires careful patient selection in order to minimize financial costs and potential toxicities. Although tissue biopsies remain the gold standard for tumor genotyping, liquid biopsies have emerged as potential new tools for clinicians to guide treatment strategies in clinical practice.

Liquid biopsy could be very useful in better identifying patients most likely to benefit from immunotherapy. Despite several blood biomarkers having been evaluated, we are far from the identification of a reliable predictive biomarker. It is possible that a single biomarker could not be able to predict response to immunotherapy due to the heterogeneous nature of cancer and the complex interaction between the tumor and TME. Sequencing technologies are rapidly evolving, but the standardization of complete diagnostic workflows to ensure high reproducibility of the results will be necessary for broader use of liquid biopsy. Further prospective studies are required to validate the routine use of liquid biopsy in NSCLC patients receiving ICIs.

## Figures and Tables

**Table 1 cancers-13-01794-t001:** Summary of the main evidences from trials on liquid biopsy in non-small cell lung cancer (NSCLC) patients receiving immunotherapy.

Method	Authors	Sample Size	ICI(s)	Cutoff(s)	Results	Annotations
ctDNA	Gandara, et al. [20]	*n* = 853	atezolizumab	TMB ≥ 16	bTMB indipendently predicts PFS benefit PFS outcomes best in pts with combined high bTMB and PD-L1 expression	tTMB and bTMB are positively correlated bTMB is not associated with high PD-L1 expression
	Wang et al. [21]	*n* = 98	anti-PD1/PD-L1	TMB > 6	High bTMB associates with better PFS and ORR	This study validates the NCC-GP150 panel for ctDNA-based bTMB measure
	Goldberg et al. [22]	*n* = 28	anti-PD1/PD-L1	ctDNA drop ≥ 50% from baseline	ctDNA response associates with superior PFS and OS	ctDNA is an early marker of clinical benefit Pts with ctDNA responses are more likely to have longer duration of treatment benefit
	Ijima et al. [23]	*n* = 14	nivolumab	VAF ≥ 2%	Early (i.e., first 2 weeks) changes in ctDNA levels predict treatment benefit	ctDNA is detected more frequently in pts with high tumor burden
	Raja et al. [24]	*n* = 73	durvalumab	dVAF < 0	dVAF associates with ORR, longer DOR and improved PFS and OS dVAF is an early marker for clinical benefit	dVAF does not significantly correlate with PD-L1 status
	Guibert et al. [25]	*n* = 86	nivolumab, pembrolizumab	dVAF < 30% and 50%	Pts with any decrease of ctDNA AF at one month have longer PFS and increased DOR	“High immune score” associated with better PFS PD-L1 expression less predictive of response than ctDNA profiling
	Socinski et al. [26]	*n* = 152	atezolizumab	TMB ≥ 16	High bTMB has numerical benefit for PFS and OS	Decreased serum CRP over 6 weeks predicts PFS and OS benefit
CTCs	Tamminga et al. [27]	*n* = 63	anti-PD1/PD-L1, anti-PD1 + anti-CTLA4	CTC ≥ 1 tdEV ≥ 18	CTC is an independent predictive factor for durable tumor response rates	tdEV are not associated with response, but with worse PFS and OS
	Castello et al. [28]	*n* = 35	anti-PD1/PD-L1, anti-PD1 + anti-CTLA4	CTC ≥ 1	CTC count at 8 weeks is an independent predictor for PFS and OS Combination of mean CTC and median MTV at 8 weeks associates with PFS and OS	CTC correlates with tumor burden CTC count variation is associated with tumor metabolic response at 18FDG-PET/CT scan
	Nicolazzo et al. [29]	*n* = 24	nivolumab	PD-L1(+) CTC ≥ 1	PD-L1(+) CTCs at 6 months of treatment correlates with PD	Presence of CTCs and expression of PD-L1 are associated with poor outcomes
	Guibert et al. [30]	*n* = 96	nivolumab	PD-L1(+) CTC ≥ 1	Higher baseline PD-L1(+) CTC associates with poor PFS	No correlation between tissue and CTC PD-L1 expression
	Dhar et al. [31]	*n* = 22	nivolumab, avelumab	>1.32 CTCs/mL PD-L1(+) CTC ≥ 2	High PD-L1(+) CTC associates with better DCR	Combination of PD-L1 positivity on tumor tissue and on CTCs as a potential biomarker for clinical benefit
	Kulasinghe et al. [32]	*n* = 33	anti-PD1/PD-L1	>1 CTC/3.75 mL	Presence of CTCs at baseline is not associated with PFS, nor is CTC PD-L1 expression status	Different role of CTCs in HNC and NSCLC
miRNAs/exosomes	Del Re et al. [33]	*n* = 8	nivolumab, pembrolizumab	PD-L1 miRNA copies	Exosomal PD-L1 expression associates with response	Concordant results among pts with NSCLC and melanoma
	Halvorsen et al. [34]	*n* = 20	nivolumab	NA	7miRNA signature associates with survival	Validation in an independent cohort is needed
	Costantini et al. [35]	*n* = 43	nivolumab	sPD-L1 > 33.97 pg/mL	High sPD-L1 and its increase associates with worse PFS and OS Pts with clinical benefit had a down expression in miRNA-320b and -375	sPD-L2, sIL-2 and sIFN-γ associate with serious irAEs

Abbreviations: 18FDG-PET/CT, 18f-fluorodeoxyglucose positron emission tomography/computed tomography scan; bTMB, blood tumor mutational burden; CRP, C-reactive protein; CTC, circulating tumor cells; ctDNA, circulating tumor DNA; CTLA-4, cytotoxic T-lymphocyte antigen 4; DCR, disease control rate; DOR, duration of response; dVAF, change in mean variant allelic frequency; HNC, head and neck cancer; ICIa, immune checkpoint inhibitors; irAEs, immune-related adverse events; miRNAs, microRNAs; MTV, metabolic tumor volume; NA, not applicable; NSCLC, non-small cell lung cancer; ORR, objective response rate; OS, overall survival; PD, progressive disease; PD-1, programmed cell death 1; PD-L1, programmed cell death ligand 1; PD-L2, programmed cell death ligand 2; PFS, progression free survival; pts, patients; sIFN-, soluble interferon gamma; sIL-s, soluble interleukin 2; sPD-L1/2, soluble programmed cell death ligand 1/2; tdEV, tumor-derived extracellular vesicles; VAF, variant allelic frequency.

**Table 2 cancers-13-01794-t002:** Overview of ongoing clinical trials of liquid biopsy techniques in NSCLC undergoing immunotherapy.

Clinical Trial Name, Number	Condition(s)	Sample Size	Assessments	Aim of Liquid Biopsy Analysis
TRACELib002, NCT04566432	Treatment naïve Stage IIIB or Stage IV NSCLC	*n* = 250	Tissue biopsy and ctDNA liquid biopsy	the evolution of ctDNA mutation profile during treatment
NCT04490564	HNSCC, NSCLC, or melanoma assigned to receive PD1 inhibitor	*n* = 185	Tissue biopsy and PD-L1 CTCs	Clinical performance of PD-L1 kit in CTCs of peripheral blood and tumor tissue samples
NCT03512847	Metastatic NSCLC assigned for immunotherapy or chemotherapy	*n* = 150	Tissue biopsy and longitudinal blood samples for ctDNA	Predictive gene profiles; resistance mechanisms toward chemotherapy and immunotherapy; ctDNA as a dynamic biomarker
NCT04636047	Immunotherapy- naïve NSCLC	*n* = 450	Blood samples analyzed by NGS CGP panel for mutations with sensitivity/resistance to targeted therapies, bTMB, HLA.	PFS, bTMB
K20-188, NCT04372732	NSCLC PD-L1 expression ≥1% before anti-PD1 treatment	*n* = 200	Tumor autoantibodies on blood samples	Correlation of tumor autoantibodies and PFS/ORR of PD-1 blockade treatment.
NCT03360630 *	Advanced NSCLC treated with anti-PD1 with or without DC-CIK immunotherapy	*n* = 60	ctDNA on blood samples	Correlations of ctDNA with clinical outcomes *

Abbreviations: bTMB, blood tumor mutational burden; CTC, circulating tumor cells; ctDNA, circulating tumor DNA; DC-CIK, dendritic cell-cytokine induced killer cell; HLA, human leukocyte antigen; HNSCC, head and neck squamous cell carcinoma; NGS CGP, next generation sequencing comprehensive genomic profile; NSCLC, non-small cell lung cancer; ORR, objective response rate; PD-1, programmed cell death protein 1; PD-L1, programmed cell death protein ligand 1; pExo, plasma exosomes; PFS, progression free survival. * randomized clinical trial o compared the clinical effects and safety of immunotherapy with dendritic cells and cytokine-induced killer cells administered with anti-PD-1 antibody in advanced NSCLC; among study’s outcome measures: to investigate the relationship of ctDNA with clinical outcomes.

## Data Availability

Not applicable.

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
