# Peer review of "Circulating Biomarkers of Response and Toxicity of Immunotherapy in Advanced Non-Small Cell Lung Cancer (NSCLC): A Comprehensive Review"

_cancers, 2021, doi:10.3390/cancers13081794_

Round 1
Reviewer 1 Report
In this review, the authors have assessed the existing literature to provide an overview of the use of liquid biopsies in the immunotherapy setting. Both immunotherapy response and the development of adverse events in NSCLC patients have been reviewed. (Dis)advantages of using liquid biopsies (e.g. cfDNA, exosomes, miRNA and others) and implications for future use have been reported. The review provides a good overview of the work that has been performed in the field. I have the following comments to make:
- Although, in general, the cost of high throughput sequencing techniques is a problem, I don’t see how the cost of liquid biopsies (compared to tissue biopsies) provide a challenge (sentence 72)?
- The authors mention that several studies have shown that irAEs can be indicative of better response to immunotherapy. There are, however, studies that indicate the opposite. This is strongly dependent on the type and severity of the irAEs that are refered to (rash versus checkpoint inhibitor pneumonitis for example). A more elaborate discussion on this point is necessary. Some references:
Suresh K, Psoter KJ, Voong KR, Shankar B, Forde PM, Ettinger DS, et al. Impact of Checkpoint Inhibitor Pneumonitis on Survival in NSCLC Patients Receiving Immune Checkpoint Immunotherapy. J Thorac Oncol. 2019;14(3):494-502.
Haratani K, Hayashi H, Chiba Y, Kudo K, Yonesaka K, Kato R, et al. Association of Immune-Related Adverse Events With Nivolumab Efficacy in Non-Small-Cell Lung Cancer. JAMA Oncol. 2018;4(3):374-8.
Reuss JE, Suresh K, Naidoo J. Checkpoint Inhibitor Pneumonitis: Mechanisms, Characteristics, Management Strategies, and Beyond. Curr Oncol Rep. 2020;22(6):56.
Author Response
Reviewer #1
In this review, the authors have assessed the existing literature to provide an overview of the use of liquid biopsies in the immunotherapy setting. Both immunotherapy response and the development of adverse events in NSCLC patients have been reviewed. (Dis)advantages of using liquid biopsies (e.g. cfDNA, exosomes, miRNA and others) and implications for future use have been reported. The review provides a good overview of the work that has been performed in the field. I have the following comments to make:
Comment: Although, in general, the cost of high throughput sequencing techniques is a problem, I don’t see how the cost of liquid biopsies (compared to tissue biopsies) provide a challenge (sentence 72)?
Response: We thank the reviewer for this correction, and corrected the sentence as suggested, removing “costs” as potential challenges of liquid biopsy (Line 80).
Comment: The authors mention that several studies have shown that irAEs can be indicative of better response to immunotherapy. There are, however, studies that indicate the opposite. This is strongly dependent on the type and severity of the irAEs that are refered to (rash versus checkpoint inhibitor pneumonitis for example). A more elaborate discussion on this point is necessary. Some references:
Suresh K, Psoter KJ, Voong KR, Shankar B, Forde PM, Ettinger DS, et al. Impact of Checkpoint Inhibitor Pneumonitis on Survival in NSCLC Patients Receiving Immune Checkpoint Immunotherapy. J Thorac Oncol. 2019;14(3):494-502.
Haratani K, Hayashi H, Chiba Y, Kudo K, Yonesaka K, Kato R, et al. Association of Immune-Related Adverse Events With Nivolumab Efficacy in Non-Small-Cell Lung Cancer. JAMA Oncol. 2018;4(3):374-8.
Reuss JE, Suresh K, Naidoo J. Checkpoint Inhibitor Pneumonitis: Mechanisms, Characteristics, Management Strategies, and Beyond. Curr Oncol Rep. 2020;22(6):56.
Reponse: in order to comply with the reviewer’s suggestion, we elaborate the discussion on this point (Lines 72-77) and added the suggested references.
Reviewer 2 Report
This is a comprehensive and well-written review about blood biomarkers of immunotherapy in NSCLC. I would suggest adding at least 1 Figure and 1 Table that would summarize the content.
Author Response
Reviewer #2.
Comment: This is a comprehensive and well-written review about blood biomarkers of immunotherapy in NSCLC. I would suggest adding at least 1 Figure and 1 Table that would summarize the content.
Response: In order to comply with the reviewer’s suggestion, we added Table 1 reporting a summary of the main evidences from trials on liquid biopsy in NSCLC patients receiving immunotherapy; and Table 2 reporting an overview of clinical trials of liquid biopsy techniques in NSCLC undergoing immunotherapy. We have summarized the contents of the review in the graphical abstract and therefore did not add a Figure to the manuscript.
Reviewer 3 Report
The current study by Indini et al. entitled "Circulating biomarkers of response and toxicity of immuno-therapy in advanced non-small cell lung cancer (NSCLC): a comprehensive review" is an interesting review paper focusing on the available evidence regarding the possible role of liquid biomarkers in the selection of NSCLC patients who could respond to immunotherapy and the identification of patients who are most likely to experience irAEs when they are treated with immune checkpoint inhibitors. This topic has great importance since the finding of clinically useful biomarkers in this clinical setting remains an unmet need. Even though the authors have used a significant number of the published studies, there are many significant issues with this manuscript, including major issues with presentation, organization, interpretation of the results as well as the lack of significant elements of the study such as tables. Generally, the authors should present the results from the published studies in a more creative way providing comprehensive figures and tables which will help the readers of the journal.
Some points which need to be treated are:
- Authors names, affiliation etc are absent from the first page.
- Presented data on irAEs are limited. More studies on this issue should be presented.
- line 16” Please replace few with another phrase since “few” doesn’t reflect
- line 17: please replace “experienced” with experience.
- lines 38-39: Regarding HPD, please, mention in this sentence that it has been observed after administration of ICIs.
- Lines 99-106: please provide a reference.
- Lines 151-155: please provide a reference.
- Line 170: please, replace epCAM with EpCAM.
- Line 213: Please, remove “somehow”.
- Line 205-231: The section entitled “Circulating microRNAs and exosomes” needs to be rewritten and to be enriched with further studies.
- Lines 224-225: Please clarify the phrase “see further section”.
- The section “6. Peripheral blood cells” should be restructured and provided studies should be presented in paragraphs.
- The section “7. Conclusions” should be restructured. The lines 316-317 should be moved to the introduction section. The mentioned tables are not presented in the current version of the study.
Author Response
Reviewer #3.
The current study by Indini et al. entitled "Circulating biomarkers of response and toxicity of immuno-therapy in advanced non-small cell lung cancer (NSCLC): a comprehensive review" is an interesting review paper focusing on the available evidence regarding the possible role of liquid biomarkers in the selection of NSCLC patients who could respond to immunotherapy and the identification of patients who are most likely to experience irAEs when they are treated with immune checkpoint inhibitors. This topic has great importance since the finding of clinically useful biomarkers in this clinical setting remains an unmet need.
Comment: Even though the authors have used a significant number of the published studies, there are many significant issues with this manuscript, including major issues with presentation, organization, interpretation of the results as well as the lack of significant elements of the study such as tables. Generally, the authors should present the results from the published studies in a more creative way providing comprehensive figures and tables which will help the readers of the journal.
Response: We thank the reviewer for this comment. Indeed, the previously submitted version of our manuscript missed the two tables, which we have added to the present submission. Table 1 reports a summary of the main evidences from trials on liquid biopsy in NSCLC patients receiving immunotherapy; and Table 2 reports an overview of clinical trials of liquid biopsy techniques in NSCLC undergoing immunotherapy. We hope these tables help the interpretation of presented results and provide more details on ongoing research in this field.
Comment: Some points which need to be treated are: authors names, affiliation etc are absent from the first page.
Response: We apologize for this mistake, and added the author’s names with affiliation and contact details of the corresponding author to the first page.
Comment: Presented data on irAEs are limited. More studies on this issue should be presented.
Response: In order to comply with the reviewer’s comment, we elaborated the issue of irAEs in the Introduction section (Lines 71-77). However, data on the correlation of liquid biopsy and irAEs onset are limited, as suggested by the results of studies presented in the Tables. We performed another thorough research of the Literature (PubMed and Scopus) searching for studies reporting data on this association, but did not find adjunctive research to report.
Comment: line 16” Please replace few with another phrase since “few” doesn’t reflect.
Response: in order to comply with the reviewer’s request, we corrected “few” with “a small portion of patients” (Lines 23-24).
Comment: line 17: please replace “experienced” with experience.
Response: thank you for the correction, we corrected this typo.
Comment: lines 38-39: Regarding HPD, please, mention in this sentence that it has been observed after administration of ICIs.
Response: in order to comply with the reviewer’s comment, we added the requested comment (Lines 48-49).
Comment: Lines 99-106: please provide a reference.
Response: We thank the reviewer for this comment, and added reference 28 accordingly.
Comment: Lines 151-155: please provide a reference.
Response: in order to comply with the reviewer’s request, we added reference 37.
Comment: Line 170: please, replace epCAM with EpCAM.
Response: in order to comply with the reviewer’s request, we corrected the typo.
Comment: Line 213: Please, remove “somehow”.
Response: in order to comply with the reviewer’s request, we removed the word as suggested.
Comment: Line 205-231: The section entitled “Circulating microRNAs and exosomes” needs to be rewritten and to be enriched with further studies.
Response: In order to comply with the reviewer’s suggestion, we enriched the section entitled “Circulating microRNAs and exosomes” (Lines 227-280).
Comment: Lines 224-225: Please clarify the phrase “see further section”.
Response: in order to comply with the reviewer’s suggestion, we removed this phrase as it was generic and potentially confounding.
Comment: The section “6. Peripheral blood cells” should be restructured and provided studies should be presented in paragraphs.
Response: in order to comply with the reviewer’s suggestion, we modified section 6 “Peripheral blood cells” and divided it in paragraphs.
Comment: The section “7. Conclusions” should be restructured. The lines 316-317 should be moved to the introduction section. The mentioned tables are not presented in the current version of the study.
Response: We thank the reviewer for this important observation. As suggested, we modified the Conclusions section, moved the suggested lines into the Introduction section, and added the two tables.
Round 2
Reviewer 3 Report
In the revised version (cancers-1133025-peer-review-v2) of the study entitled "Circulating biomarkers of response and toxicity of immuno-therapy in advanced non-small cell lung cancer (NSCLC): a comprehensive review" by Indini et al., the authors have addressed the majority of the points that we had noted. However, there are the following issues that they need to be further clarified.
- High Turnitin score (about 30 %) regarding plagiarism. Please, this is important and it needs to be confirmed and treated.
- Lines 345-349: In this paragraph, NK cells are characterized as immunosuppressive cells as well as the authors mention that their “circulating fraction exert a negative effect on anti-PD1 therapy in animal models”. Both of them need to be further explained since the provided reference doesn’t mention any relevant. Please, clarify this point.
Author Response
In the revised version (cancers-1133025-peer-review-v2) of the study entitled "Circulating biomarkers of response and toxicity of immuno-therapy in advanced non-small cell lung cancer (NSCLC): a comprehensive review" by Indini et al., the authors have addressed the majority of the points that we had noted. However, there are the following issues that they need to be further clarified.
Comment: High Turnitin score (about 30 %) regarding plagiarism. Please, this is important and it needs to be confirmed and treated.
Response: We apologize for this inconvenience and thoroughly modified the manuscript accordingly. If this problem still persists, please do not hesitate to indicate us the specific section in which plagiarism has been detected in order to modify them if necessary.
Comment: Lines 345-349: In this paragraph, NK cells are characterized as immunosuppressive cells as well as the authors mention that their “circulating fraction exert a negative effect on anti-PD1 therapy in animal models”. Both of them need to be further explained since the provided reference doesn’t mention any relevant. Please, clarify this point.
Response: in order to comply with the reviewer’s comment, we discussed this point (Lines 494-498) and added reference 74, accordingly.
Round 3
Reviewer 3 Report
The similarity score remains high (24%). Especially, the CTCs section needs further evaluation regarding this issue.
The rest comments have been adressed.
Author Response
We apologies for this inconvenience and further modified the whole manuscript (and specifically the section regarding CTCs) accordingly.
We hope we have resolved this issue in this final version.
